# Crl activates transcription by stabilizing active conformation of the master stress transcription initiation factor

Juncao Xu[1,2†], Kaijie Cui[2,3†], Liqiang Shen[1,2], Jing Shi[4,5], Lingting Li[1,2], Linlin You[1,2], Chengli Fang[1,2], Guoping Zhao[1,6,7,8,9*], Yu Feng[4,5*], Bei Yang[3*], Yu Zhang[1*]

[1]Key Laboratory of Synthetic Biology,CAS Center for Excellence in Molecular Plant Sciences, Shanghai Institute of Plant Physiology and Ecology, Chinese Academy of Sciences, Shanghai, China; [2]University of Chinese Academy of Sciences, Beijing, China; [3]Shanghai Institute for Advanced Immunochemical Studies, ShanghaiTech University, Shanghai, China; [4]Department of Biophysics, Zhejiang University School of Medicine, Hangzhou, China; [5]Department of Pathology of Sir Run Run Shaw Hospital, Zhejiang University School of Medicine, Hangzhou, China; [6]Shanghai-MOST Key Laboratory of Health and Disease Genomics, Chinese National Human Genome Center at Shanghai, Shanghai, China; [7]Department of Microbiology, Li Ka Shing Institute of Health Sciences, The Chinese University of Hong Kong, Prince of Wales Hospital, Shatin, China; [8]State Key Laboratory of Genetic Engineering, Department of Microbiology, School of Life Sciences, Fudan University, Shanghai, China; [9]Institute of Biomedical Sciences, Fudan University, Shanghai, China

*For correspondence:
gpzhao@sibs.ac.cn (GZ);
yufengjay@zju.edu.cn (YF);
yangbei@shanghaitech.edu.cn (BY);
yzhang@sippe.ac.cn (YZ)

†These authors contributed equally to this work

Competing interests: The authors declare that no competing interests exist.

**Abstract** $\sigma^S$ is a master transcription initiation factor that protects bacterial cells from various harmful environmental stresses including antibiotic pressure. Although its mechanism remains unclear, it is known that full activation of $\sigma^S$-mediated transcription requires a $\sigma^S$-specific activator, Crl. In this study, we determined a 3.80 Å cryo-EM structure of an *Escherichia coli* transcription activation complex (*E. coli* Crl-TAC) comprising *E. coli* $\sigma^S$-RNA polymerase ($\sigma^S$-RNAP) holoenzyme, Crl, and a nucleic-acid scaffold. The structure reveals that Crl interacts with domain 2 of $\sigma^S$ ($\sigma^S_2$) and the RNAP core enzyme, but does not contact promoter DNA. Results from subsequent hydrogen-deuterium exchange mass spectrometry (HDX-MS) indicate that Crl stabilizes key structural motifs within $\sigma^S_2$ to promote the assembly of the $\sigma^S$-RNAP holoenzyme and also to facilitate formation of an RNA polymerase–promoter DNA open complex (RPo). Our study demonstrates a unique DNA contact-independent mechanism of transcription activation, thereby defining a previously unrecognized mode of transcription activation in cells.

## Introduction

Bacterial cells are capable of rapidly adapting to different ecological conditions through highly regulated dynamic switching among different gene-expression programs, and they do so by selectively activating distinct σ-RNA polymerase (σ-RNAP) holoenzymes (*Österberg et al., 2011*). The alternative initiating σ factor, $\sigma^S$ (also known as $\sigma^{38}$ in *Escherichia coli*) is the master stress regulator that protects many Gram-negative bacteria from detrimental environmental conditions (*Lange and Hengge-Aronis, 1991*). It also plays indispensable roles in the biofilm formation, virulence, antibiotic tolerance, and antibiotic persistence of human pathogens including *Salmonella enterica*, *Pseudomonas aeruginosa*, and *E. coli* (*Hansen et al., 2008*; *Murakami et al., 2005*; *Stewart et al., 2015*; *Wu et al., 2015*). Different stress conditions, including antibiotic treatment, could strongly induce

the expression of σ$^S$ (*Battesti et al., 2011*), which in turn activates the transcription of ~10% of genes from the *E. coli* genome by the σ$^S$-RNAP holoenzyme, thereby rendering bacterial cells resistant to antibiotic treatment and other stresses such as hydrogen peroxide, high temperature, low pH, osmotic shock and so on (*Battesti et al., 2011*; *Lelong et al., 2007*; *Weber et al., 2005*).

σ$^S$ is a group-2 alternative σ factor (*Feklístov et al., 2014*). The conserved domains of σ$^S$ (σ$^S_{1.2}$, σ$^S_2$, σ$^S_{3.1}$, σ$^S_{3.2}$, and σ$^S_4$) interact with the RNAP core enzyme through exactly the same interfaces as those of housekeeping σ factor (σ$^{70}$ in *E. coli*) (*Liu et al., 2016*). σ$^S$ shares a high degree of sequence similarity with σ$^{70}$ and prefers similar sequences at the −10 (TATAAT) and −35 (TTGACA) promoter elements (*Gaal et al., 2001*). Although σ$^S$- and σ$^{70}$-regulated genes overlap to some extent, σ$^S$ exhibits good selectivity towards its own regulon partially through a 'derivation-from-the-optimum' strategy. The σ$^S$-RNAP holoenzyme tolerates degenerate promoter sequences (mostly at the −35 element) better than σ$^{70}$-RNAP does, though at the cost of reduced overall transcription activity (*Gaal et al., 2001*; *Maciag et al., 2011*; *Typas et al., 2007b*).

Besides the inferior transcriptional activity of σ$^S$-RNAP when compared to σ$^{70}$-RNAP, the amount of *E. coli* σ$^S$ in cells is also smaller than that of σ$^{70}$ in stationary phase and stress conditions (*Jishage et al., 1996*), and the affinity of σ$^S$ is ~15 times lower than that of σ$^{70}$ to the RNAP core enzyme (*Maeda et al., 2000*). Therefore, σ$^S$ has to cooperate with its allies to compete with σ$^{70}$ for RNAP core enzyme in order to transcribe its own regulon. A large collection of genetic and biochemical data has highlighted the importance of Crl in σ$^S$-mediated transcription in bacterial cells (*Cavaliere and Norel, 2016*). Crl was demonstrated to activate σ$^S$-mediate transcription directly both in vitro and in vivo (*Banta et al., 2013*; *Banta et al., 2014*; *Cavaliere et al., 2014*; *Cavaliere et al., 2015*; *England et al., 2008*; *Monteil et al., 2010a*; *Pratt and Silhavy, 1998*; *Typas et al., 2007a*), and *Crl*-null *Salmonella* and *E. coli* cells displayed impaired biogenesis of curli (which is important for host cell adhesion and invasion as well as for formation of biofilm), increased sensitivity to H$_2$O$_2$ stress, and reduced virulence due to decreased expression of several σ$^S$-regulated genes (*Arnqvist et al., 1992*; *Barnhart and Chapman, 2006*; *Monteil et al., 2010a*; *Robbe-Saule et al., 2006*; *Robbe-Saule et al., 2008*).

Crl is a unique transcription activator in bacteria: 1) unlike other canonical bacterial transcription factors that regulate the activity of housekeeping σ factor (*Browning and Busby, 2016*), Crl shows highly stringent specificity to σ$^S$ (*Banta et al., 2013*; *Bougdour et al., 2004*); 2) Crl broadly activates σ$^S$-mediated transcription in a promoter sequence-independent manner (*Lelong et al., 2007*; *Robbe-Saule et al., 2006*; *Robbe-Saule et al., 2007*); and 3) Crl seems to act in at least two stages to boost σ$^S$-mediated transcription, namely the stage of σ$^S$-RNAP holoenzyme assembly and the stage of RPo formation (*Banta et al., 2013*; *Bougdour et al., 2004*; *England et al., 2008*). Crl has been demonstrated to interact with σ$^S_2$ and probably also with the RNAP core enzyme (*England et al., 2008*), but whether or how it interacts with DNA remains elusive. Although crystal and nuclear magnetic resonance (NMR) structures of Crl are available (*Banta et al., 2014*; *Cavaliere et al., 2014*; *Cavaliere et al., 2015*), it is still unclear how Crl interacts with σ$^S$-RNAP holoenzyme and how such interaction contributes to the transcription activation of σ$^S$-RNAP.

In this study, we determined a 3.80 Å cryo-EM structure of the transcription activation complex of Crl (*E. coli* Crl-TAC) comprising *E. coli* σ$^S$-RNAP holoenzyme, Crl, and a nucleic-acid scaffold mimicking the transcription initiation bubble. In the structure, Crl shields a large solvent-exposed surface of σ$^S_2$, and bridges σ$^S_2$ and the RNAP-β' subunit, but makes no contact with promoter DNA. The cryo-EM structure together with results of hydrogen deuterium exchange mass spectrometry (HDX-MS) and mutational studies have converged on a model in which Crl activates the σ$^S$-RNAP holoenzyme by stabilizing the active conformation of σ$^S$, thereby promoting the interaction between σ$^S$ and RNAP or DNA.

## Results

### The cryo-EM structure of *E. coli* Crl-TAC

To understand the structural basis for transcription activation by Crl, we reconstituted the Crl–TAC complex that includes the *E. coli* σ$^S$-RNAP holoenzyme, Crl, and a nucleic-acid scaffold comprising a 25-bp upstream dsDNA, a 13-bp downstream dsDNA, a non-complimentary transcription bubble (−11 to +2 with respect to the transcription start site at +1), and a 5-mer RNA primer (*Figure 1A*

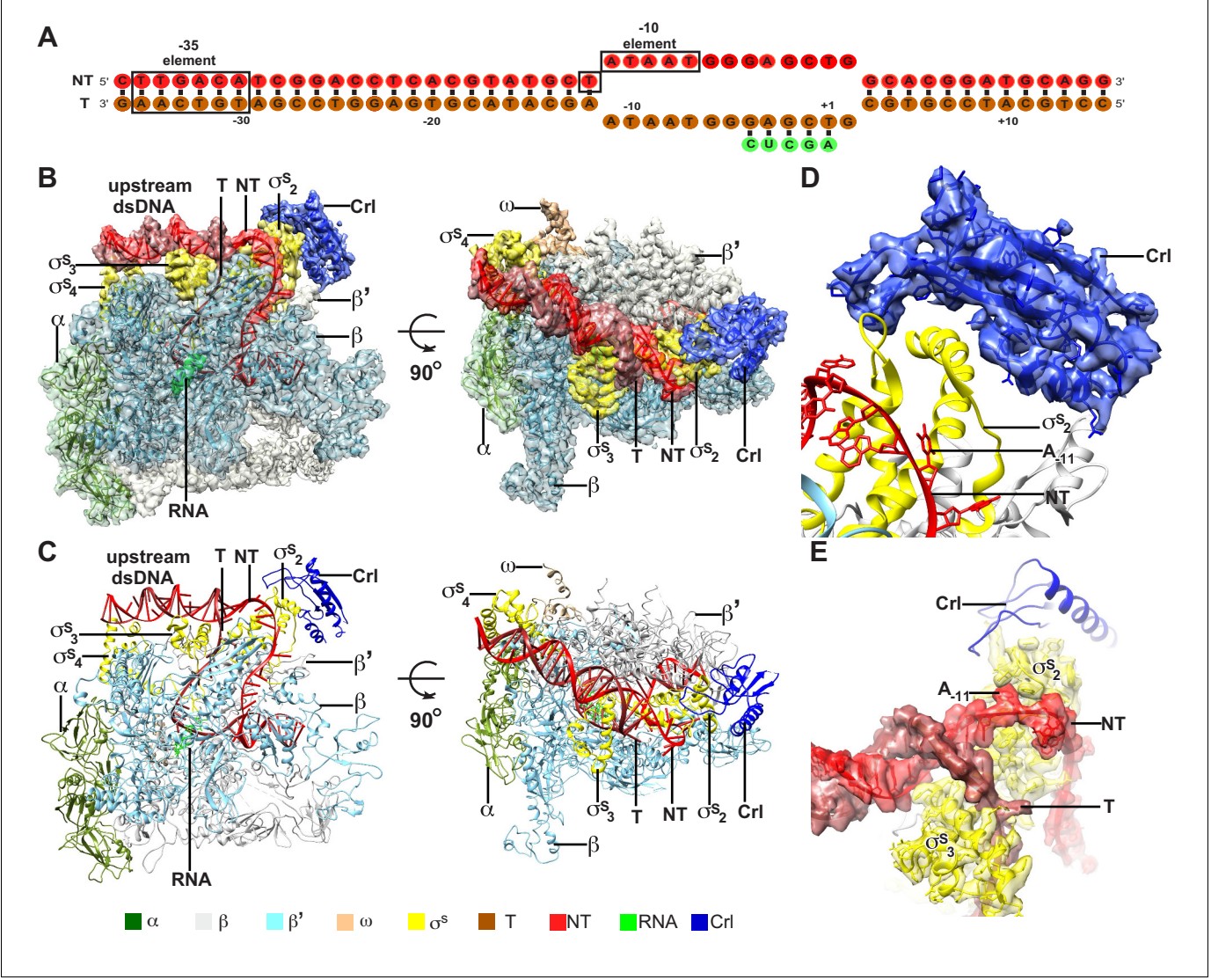

**Figure 1.** The overall structure of *E. coli* Crl–TAC. (**A**) The scaffold used in structure determination of *E. coli* Crl-TAC. (**B, C**) The top and front view orientations of the cryo-EM density map (**B**) and structure model (**C**) of *E. coli* Crl–TAC. The RNAP, Crl and nucleic acids are presented as cartoon and colored as indicated in the color key. The density map is shown in gray envelop. (**D**) The cryo-EM density map (blue transparent surface) for Crl. (**E**) The cryo-EM density map (red transparent surface) for the upstream junction of the transcription bubble of promoter DNA and σ$^S$. NT, non-template-strand promoter DNA; T, template-strand promoter DNA.

The online version of this article includes the following figure supplement(s) for figure 1:

**Figure supplement 1.** The preparation of *E. coli* Crl–TAC complex.

**Figure supplement 2.** The processing pipeline for cryo-EM map construction of *E. coli* Crl–TAC.

and *Figure 1—figure supplement 1*). The cryo-EM structure of *E. coli* Crl-TAC was determined at a nominal resolution of 3.80 Å by a single-particle reconstitution method (*Supplementary file 2* and *Figure 1—figure supplement 2*). The density map shows clear signals for the nucleic-acid scaffold, σ$^S$, and Crl (*Figure 1B and D–E*). The crystal structure of Crl could be readily fit into the density, suggesting little conformational change of Crl upon interaction with σ$^S$-RNAP holoenzyme (*Figure 1D*) (*Banta et al., 2014*).

The cryo-EM structure clearly shows that Crl locates at the outer surface of the σ$^S$-RNAP holoenzyme (*Figure 1B–C*). It mainly interacts with σ$^S_2$ and shields the σ$^S_2$ through an interface of 695 Å$^2$; it also contacts a helical domain of RNAP-β' clamp through a very small interface of ~85 Å$^2$ (*Figure 2A–D*). Crl approaches the upstream edge of transcription bubble, but makes no direct

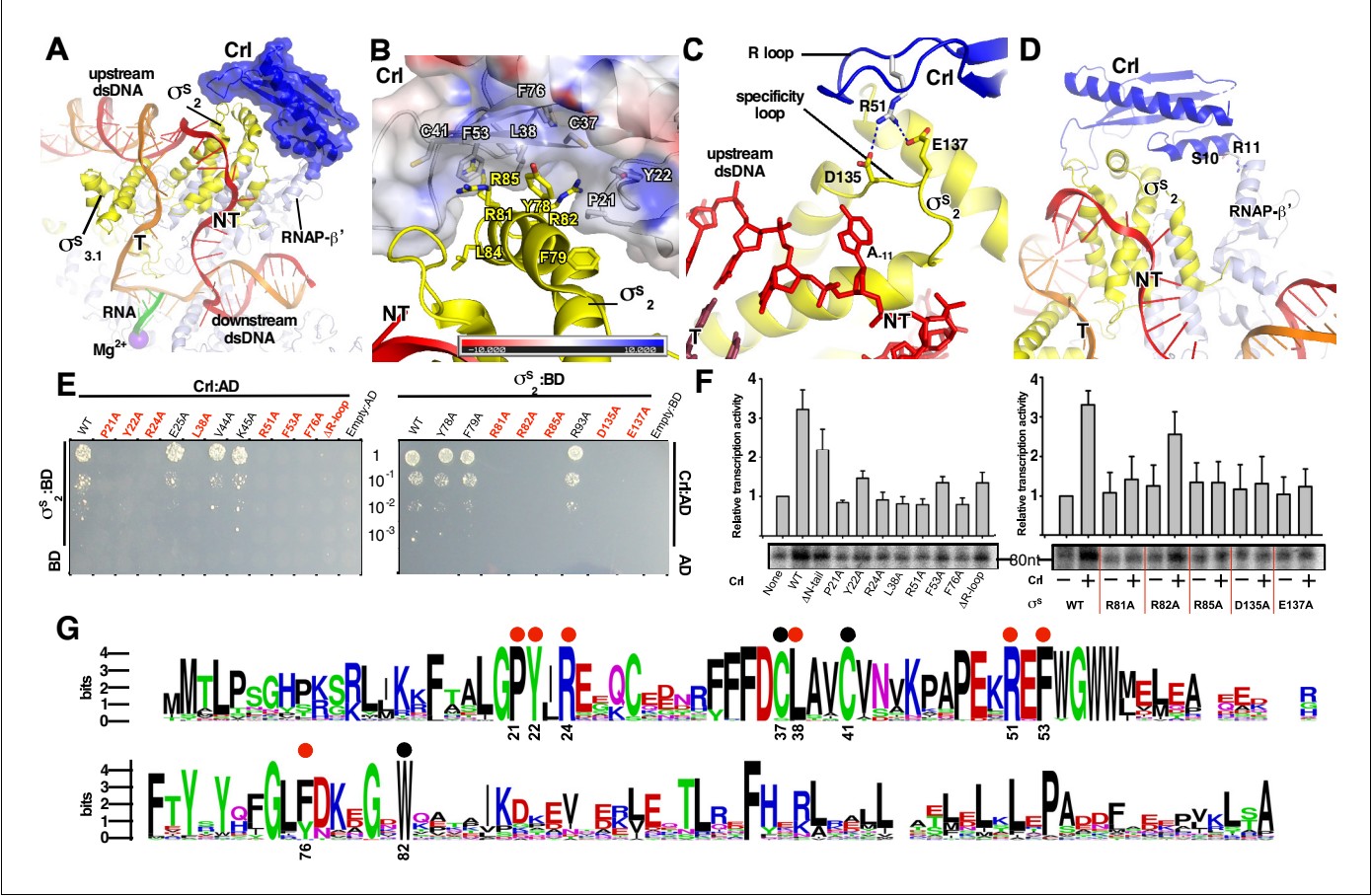

**Figure 2.** The interactions between Crl and the RNAP-holoenzyme. (**A**) Crl binds to the $\sigma^S_2$ and the RNAP-β' subunit. Crl is represented as a blue transparent surface and cartoon. (**B**) $\sigma^S$ is embedded into a shallow hydrophobic groove of Crl. The electrostatic potential surface of Crl was generated using APBS tools in Pymol. (**C**) The detailed salt-bridge interaction between the 'R' loop of Crl and the 'specificity loop' of $\sigma^S$. Salt-bridge bonds are shown as blue dashed lines. (**D**) The N-terminal tail of Crl makes potential weak interactions with RNAP-β' subunit, probably through R11 and S10. The colors are as in *Figure 1*. (**E**) The yeast two-hybrid assay reveals the key interface residues (red) of Crl (left) or $\sigma^S$ (right). Mutating the key residues disrupts interactions between Crl and $\sigma^S$. AD, the activation domain of GAL4; BD, the DNA-binding domain of GAL4. (**F**) The in vitro transcription assay shows that mutating most of the key interface residues of Crl (left) substantially impairs its transcription activation activity, and that mutating most of the key interface residues of $\sigma^S$ resulted in reduced response to Crl. ΔR-loop, replacing residue 43–51 with a 'GSGS' linker. (**G**) Protein sequence alignment of Crl from 72 non-redundant bacterial species. Filled circles indicate residues that are involved with interactions with $\sigma^S$; filled red circles indicate key contact residues. The residues are numbers as in *E. coli* Crl. NT, non-template-strand promoter DNA; T, template-strand promoter DNA.

The online version of this article includes the following figure supplement(s) for figure 2:

**Figure supplement 1.** The detailed interactions between Crl and $\sigma^S_2$.

**Figure supplement 2.** The sequence alignment of bacterial σ factors.

contact with the promoter DNA (*Figures 1D–E* and *2A*). Such a mode of interaction between Crl and $\sigma^S$-RPo supports previous findings from genetic, bacterial two-hybrid, cross-linking, surface plasmon resonance (SPR), NMR, or bioinformatic approaches (*Banta et al., 2013*; *Banta et al., 2014*; *Cavaliere et al., 2014*; *England et al., 2008*; *Monteil et al., 2010a*; *Monteil et al., 2010b*; *Pratt and Silhavy, 1998*).

## Crl interacts with $\sigma^S$ and the RNAP core enzyme

In the structure of *E. coli* Crl–TAC, a structural motif that includes helix α2 of $\sigma^S_{1.2}$ (residues 72–82), $\sigma^S_{NCR}$ (residues 83–89), and helix α3 of $\sigma^S_{2.1}$ (residues 90–93) is embedded into a shallow groove on Crl, and the nearby 'specificity loop' of $\sigma^S_{2.3}$ (residues 133–141) is further anchored by Crl (*Figure 2A–C*; *Figure 2—figure supplements 1–2*). Several Crl residues (P21, Y22, C37, L38, C41, F53, F76, and W82) create a hydrophobic patch in the shallow groove and make van der Waals

interactions with residues (Y78, F79, R81, R82, L84, and R85) of σ$^S$ (*Figure 2B* and *Figure 2—figure supplement 1B*). Moreover, potential polar interactions between Crl (R24) and σ$^S$ (D87) might also contribute to the interaction. The interface residues identified here recapitulate most of the hits in a previous genetic screen to identify interface residues of σ$^S$ and Crl (*Banta et al., 2014*). Intriguingly, most evolutionarily conserved residues of Crl are clustered in the shallow groove, implicating a functional relevance of its interaction with σ$^S$ (*Figure 2G*). Moreover, the Crl derivatives F53A and W82A rendered salmonella cells more sensitive to H$_2$O$_2$ stress (*Monteil et al., 2010a*), demonstrating the physiological importance of such an interface.

We subsequently evaluated contribution of each interface residue to σ$^S$–Crl interaction using a yeast two-hybrid assay. The results show that mutating most interface residues (P21W, Y22A, R24A, L38A, F53A, or F76A of Crl; R81A, R82A or R85A of σ$^S_2$) substantially impaired σ$^S$–Crl interactions (*Figure 2E*). Moreover, the results from an in vitro transcription assay show that alanine substitutions of R81, R82, or R85 of σ$^S$ impeded its response to Crl; and that alanine substitutions of L38 or F53 of Crl also impaired its ability to activate transcription (*Figure 2F*). The results validate our structure and confirm the significance of the σ$^S$–Crl interface for the transcription activation activity of Crl.

Previous studies suggested that a conserved 'DPE' motif within σ$^S_2$ plays an indispensable role in its interaction with Crl (*Banta et al., 2013*; *Banta et al., 2014*). The 'DPE' motif is part of the 'specificity loop' of σ$^S_2$, an essential structural element responsible for recognizing and stabilizing the unwound nucleotide at the most conserved position of promoter DNA (i.e. position −11 for σ$^S$ and σ$^{70}$-reglted promoters) in all bacterial transcription initiation complexes (*Campagne et al., 2014*; *Li et al., 2019*; *Lin et al., 2019*; *Liu et al., 2016*; *Zhang et al., 2012*). In our structure, the 'specificity loop' encloses the NT-11A nucleotide as reported (*Figures 2C* and *4D*; *Figure 2—figure supplement 1D*) (*Liu et al., 2016*). Notably, the conformation of the 'specificity loop' is secured by the 'R' loop (residues 41–53) of Crl. The side chain of Crl R51 reaches D135 and E137 of the conserved 'DPE' motif of σ$^S_2$, and probably makes salt-bridge interactions with them (*Figure 2C* and *Figure 2—figure supplement 1C*). Our results from yeast two-hybrid and in vitro transcription assays show that mutating either D135/E137 of σ$^S_2$ or R51 of Crl significantly compromised the Crl-σ$^S_2$ interaction and Crl-mediated transcription activation, highlighting the importance of such an interface (*Figure 2E–F*).

Our structure of Crl-TAC also explains the strict specificity of Crl to σ$^S$. Crl binds to two of the least conserved regions of bacterial σ factors: the σ$^S_{NCR}$ and the 'specificity loop' (*Figure 2—figure supplement 2A*). The sequence alignments of ten bacterial σ$^S$ and ten primary σ factors clearly show that most Crl-interacting residues on σ$^S$ are not present in σ$^{70}$ (*Figure 2—figure supplement 2B–C*).

In agreement with the previous SPR results (*England et al., 2008*), the density of Crl also suggests possible interactions between Crl and the RNAP-β' clamp domain (*Figure 1D*). The interface is relatively small and only involves residues S10 and R11 of the Crl N-terminal loop (*Figure 2D*). Deletion of the N-terminal loop of Crl shows only marginal effect on its activation of transcription (left panel of *Figure 2F*).

## Crl stabilizes σ$^S_2$ to facilitate the assembly of the σ$^S$-RNAP holoenzyme

It has been suggested that Crl may function as a σ$^S$ chaperon to facilitate the assembly of the σ$^S$-RNAP holoenzyme (*Banta et al., 2013*). We confirm that Crl is able to increase (~3.8 fold) the binding affinity between σ$^S$ and the RNAP core enzyme in a fluorescence polarization assay using fluorescein-labeled σ$^S$ (*Figure 3A*). Moreover, an 'R loop'-mutated derivative of Crl completely lost its effect on the assembly of the σ$^S$-RNAP holoenzyme, whereas deleting the N-terminal tail of Crl moderately affects this assembly (*Figure 3B* and *Supplementary file 2*), in agreement with our in vitro transcription results (*Figure 2E–F*). Such results further suggest that Crl facilitates the assembly of the σ$^S$-RNAP holoenzyme mainly through its interaction with σ$^S$ rather than through its interaction with the RNA-β' clamp.

A previous report showed that Crl shifts the equilibrium towards σ$^S$-RNAP holoenzyme formation by promoting the association between σ$^S$ and RNAP rather than by preventing the dissociation of the two (*England et al., 2008*). Such facts imply that Crl may stabilize σ$^S$ in a conformation that is more accessible for the RNAP core enzyme. To investigate this hypothesis, we characterized the in-solution dynamic conformations of full-length σ$^S$ using HDX-MS.

Deuterium exchange of σ$^S$ was first monitored in the absence of Crl at different time points. At the earliest time point, 10 s, low deuterium incorporation (≤50%) was observed for most peptides

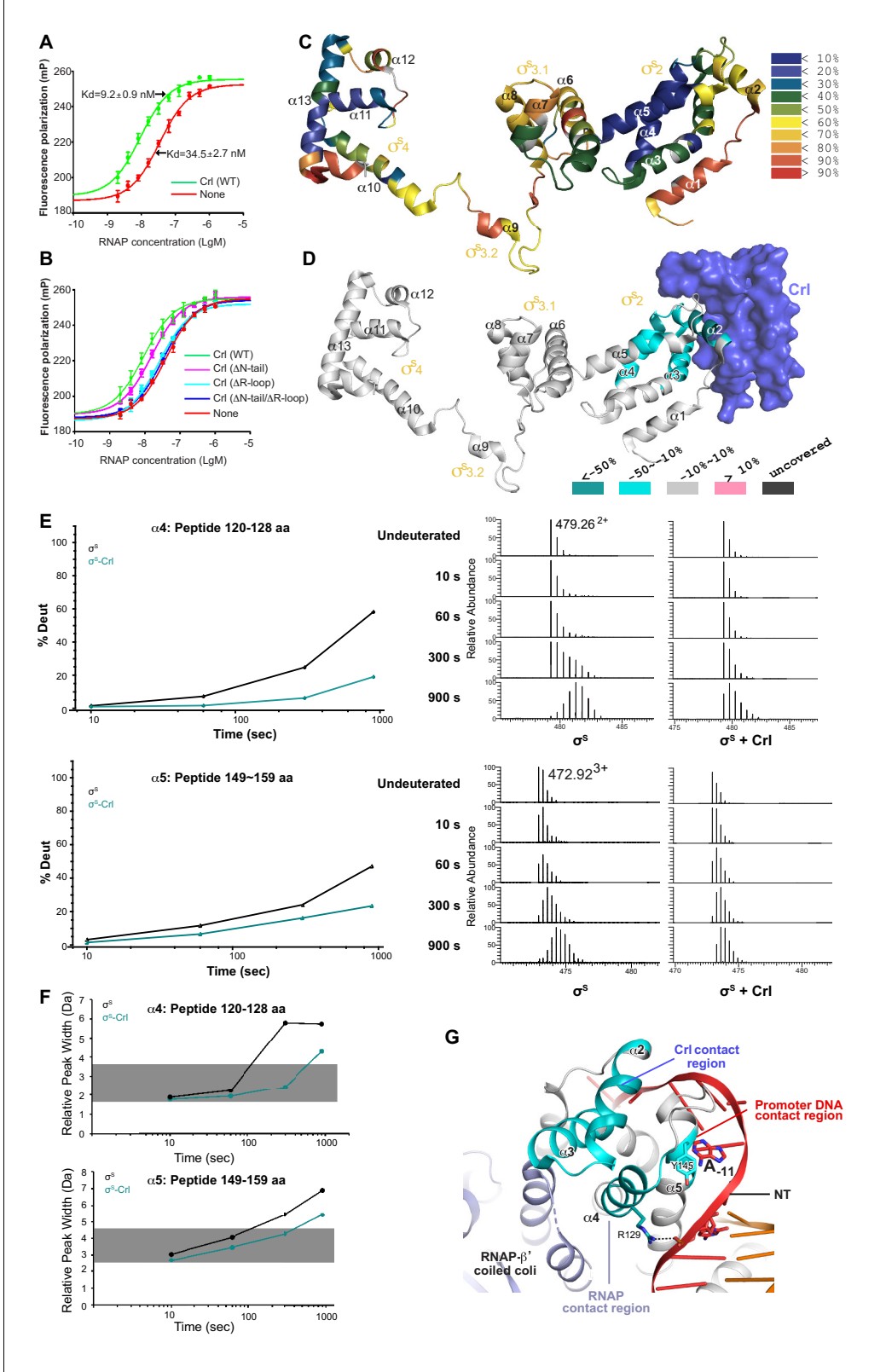

**Figure 3.** Crl promotes assembly of the $\sigma^S$-RNAP holoenzyme by stabilizing $\sigma^S_2$. (**A**) Crl increases binding affinity between RNAP and $\sigma^S$ in a fluorescence polarization (FP) assay. (**B**) Deletion the 'R' loop but not the N-tail ($\Delta$N-tail; deleting residue 1–11) abolishes the ability of Crl to promote the assembly of the $\sigma^S$-RNAP holoenzyme. (**C**) Hydrogen-to-deuterium exchange (HDX) profile of free $\sigma^S$ after 10 s exposure to deuteron. Incorporation was mapped onto the cryo-EM structure of $\sigma^S$. The inset shows the color coding for different percentages of deuteron incorporation. (**D**)

*Figure 3 continued on next page*

*Figure 3 continued*

HDX changes for σ$^S$ in the presence of Crl as compared to free σ$^S$, mapped on the cryo-EM structure of σ$^S$. The percentage difference in deuterium uptake values represents an average across all four time points, ranging from 10 s to 900 s. For heatmap color coding, pink, cyan, and gray indicate increase, decrease, and no significant change in HDX, respectively. Dark grey represents regions that were not consistently resolved in all HDX experiments. (E) Deuterium uptake plots and mass spectra of indicated peptides from helices α4 and α5 in the absence and presence of Crl. Left: the deuterium uptake data are plotted as percent deuterium uptake versus time on a logarithmic scale. Right: mass spectra of the indicated peptides at different labeling time points, with the mass spectra of undeuterated samples shown as controls. (F) Peak-width analysis at different time points for peptides from helices α4 and α5 reveal the existence of EX1 kinetics. (C–F) The HDX experiments were repeated at least twice. (G) σ$^S_2$ regions with reduced HDX rate upon Crl binding, the colors are as in (D).

The online version of this article includes the following source data and figure supplement(s) for figure 3:

**Source data 1.** The raw data of *Figure 3A-B*.
**Figure supplement 1.** HDX profile of the σ$^S$ protein in the absence of Crl.
**Figure supplement 2.** Representative peptides from helix α2, α3, α4 and α5 of σ$^S_2$ whose HDX rate decreased considerably in the presence of Crl.
**Figure supplement 3.** Deuterium uptake plots for additional peptides from σ$^S$ whose HDX rate did not change in the presence of Crl.
**Figure supplement 4.** Peptide coverage of σ$^S$.

from σ$^S_2$ and σ$^S_4$ (*Figure 3C* and *Figure 3—figure supplement 1*), and the deuterium uptake of these peptides increased gradually over time (*Figure 3—figure supplements 1–3*). Such facts indicate that these two domains generally adopt folded structures in solution, as observed in the cryo-EM structure.

Meanwhile, although σ$_{3.1}$ appears to be folded in our cryo-EM structure (helix α6–α8), most peptides from these regions (except for those ranging residues 182–197) manifest high deuterium incorporation level (≥50%) at the earliest time point (10 s) and unchanged deuterium uptake up to 900 s (*Figure 3C* and *Figure 3—figure supplements 1* and *3*), indicating that σ$_{3.1}$ is largely solvent-exposed in solution. Although domain σ$_{1.1}$ (residues 1–55) was not resolved in the cryo-EM structure, fair peptide coverage was achieved for this region in the HDX-MS experiment (*Figure 3—figure supplement 4*) and it shows rapid deuterium incorporation (*Figure 3—figure supplement 1*). Consistent with their remote locations in the cryo-EM structure, helix α1 from σ$_{1.2}$ (residues 56–67) and the σ$_{3.2}$ linker (residues 218–245) generally appear to be solvent-exposed in the HDX-MS experiment (*Figure 3C*; *Figure 3—figure supplements 1* and *3*). The HDX profile of σ$^S$ in the absence of Crl also agrees well with a recent report (*Cavaliere et al., 2018*).

We then monitored the deuterium exchange profile of σ$^S$ in the presence of Crl. In the HDX-MS results, increased protection from HDX was consistently observed for peptides ranging residues 74–99 (*Figure 3D*; *Figure 3—figure supplement 2A–B*), which span helix α2 of σ$^S_{1.2}$ (residues 74–82), σ$^S_{NCR}$ (residues 83–89), and the N-terminal part of helix α3 from σ$^S_{2.1}$ (residues 90–99). The observed protection from HDX indicates that these structural motifs from σ$^S$ may locate at the interface between σ$^S$ and Crl, which is highly consistent with our cryo-EM structure (*Figures 1D* and *2A–B*).

Intriguingly, increased protection from HDX was also observed for certain regions on α4 and α5 (*Figure 3D–E*; *Figure 3—figure supplement 2C–D*), all of which locate away from the interface between σ$^S$ and Crl as observed in the cryo-EM structure (*Figure 3D*). Hence, besides directly shielding the motifs that span α2−α3, Crl seemingly also imposes an allosteric stabilizing effect on σ$^S$. We thus analyzed the raw spectra of all peptides from these two regions. To our surprise, in the absence of Crl, typical bimodal shaped isotope clusters were observed for all peptides spanning 118–131aa and 148–161aa at longer time points (*Figure 3E*; *Figure 3—figure supplement 2C–D*). Consistently, the maximum peak widths of these peptides also increased dramatically as the labeling period was prolonged (*Figure 3F*), indicating an EX1 exchange mechanism in these regions (*Weis et al., 2006*). Together, these sources of evidence suggest that, in the absence of Crl, the aforementioned regions on α4 and α5 underwent an unfolding event that is long enough to allow hydrogen-to-deuterium exchange to happen at all exposed residues (*Wales and Engen, 2006*). Addition of Crl apparently helps to restrain these regions on σ$^S_2$ in their low HDX conformation (*Figure 3E–F*; *Figure 3—figure supplement 2C–D*), and the HDX protection effect appears to be highly pronounced for peptides from α4.

Notably, α4 is one of the major anchor point on σ$^S_2$ for the RNAP-β' subunit (*Figure 3G*). Hence, the stabilizing effect rendered on helix α4 by Crl probably facilitates the docking of σ$^S_2$ to

the RNAP-β' subunit for subsequent proper positioning of other domains onto the RNAP core enzyme, thereby promoting the assembly of functional $\sigma^S$-RNAP holoenzyme. Such a hypothesis is also supported by the increased rate of association that occurs during formation of $\sigma^S$-RNAP holoenzyme in the presence of Crl (*England et al., 2008*).

## Crl stabilizes $\sigma^S_2$ to assist in RPo formation

A previous report suggested that Crl is also able to boost $\sigma^S$-mediated transcription at a step after $\sigma^S$-RNAP holoenzyme assembly (*Bougdour et al., 2004*; *England et al., 2008*). We here show that

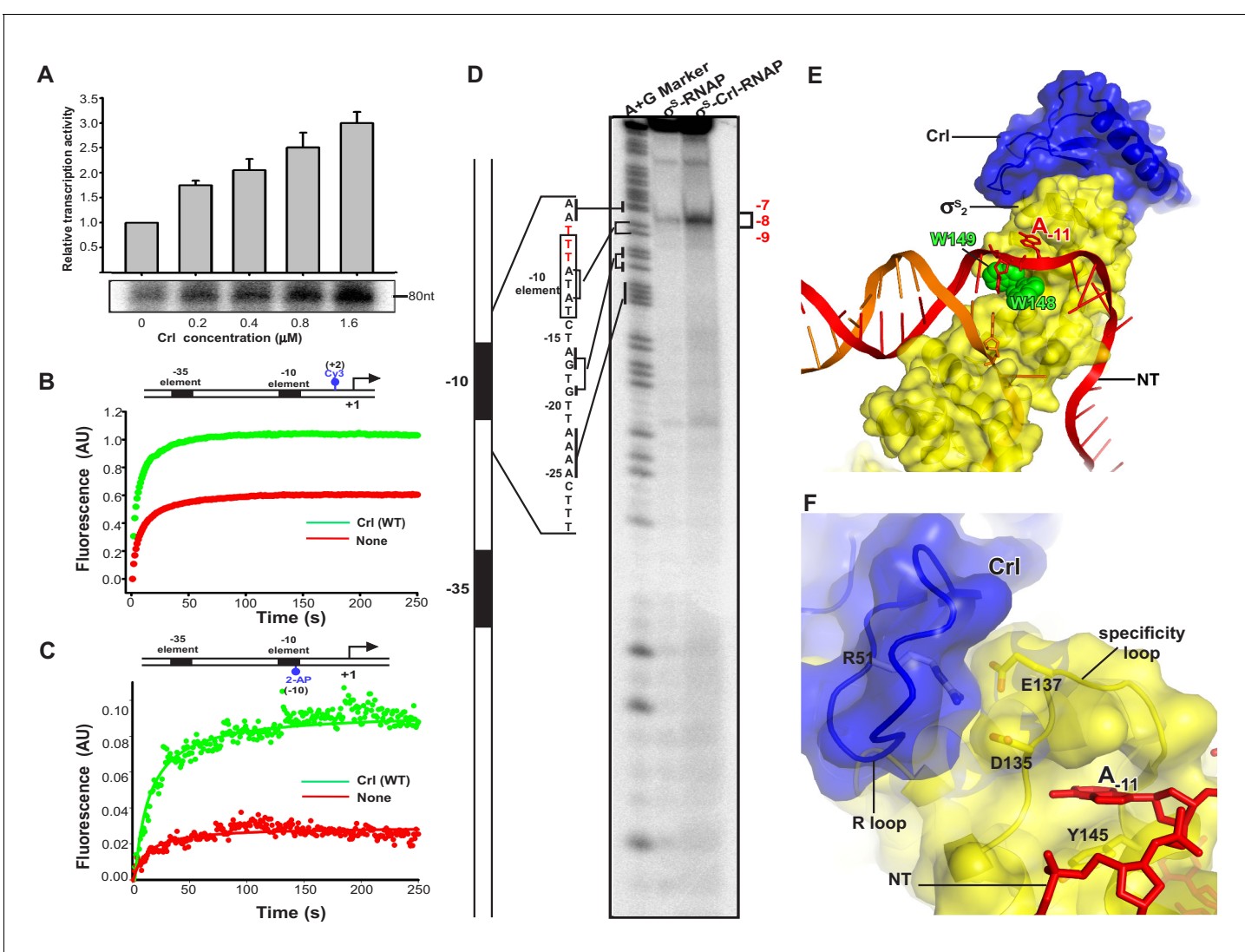

**Figure 4.** Crl facilitates RPo formation. (A) Crl increases transcription by a pre-assembled $\sigma^S$-RNAP holoenzyme from the P*osmY* promoter in a concentration-dependent manner. (B) Crl promotes $\sigma^S$-RPo formation in a fluorescence stopped-flow experiment using Cy3-labeled $\lambda P_R$ promoter DNA. (C) Crl promotes $\sigma^S$-RPo formation in a fluorescence stopped-flow experiment using 2AP-labeled $\lambda P_R$ promoter DNA. (D) The KMO$_4$ footprinting experiment result shows that Crl increases formation of the RPo complex. The *osmY* promoter dsDNA with $^{32}$P-labeled at the 5'-end of the non-template strand was used for the assay. The hyperactive T-stretch ($-9$ to $-7$) in the transcription bubble upon RPo formation is highlighted in red and labeled on the right. (E) The upstream promoter DNA is unwound by the W-dyad (W148 and W149), and the unwound A$_{-11}$ nucleotide of the non-template strand DNA is recognized and stabilized by a protein pocket on $\sigma^S_2$. (F) The 'R' loop of Crl stabilizes the conformation of the 'specificity' loop that forms the pocket for unwinding and recognizing the A$_{-11}$ nucleotide of non-template strand promoter DNA.

The online version of this article includes the following figure supplement(s) for figure 4:

**Figure supplement 1.** The promoter DNA used for stopped-flow fluorescence experiments.

Crl increases the transcription activity of a pre-assembled $\sigma^S$-RNAP holoenzyme in an in vitro transcription assay (*Figure 4A*), supporting the previous finding.

To understand how Crl activates $\sigma^S$-mediated transcription beyond the assembly of $\sigma^S$-RNAP holoenzyme, we modified a stopped-flow fluorescence assay to monitor the potential effect of Crl on RPo formation using pre-assembled $\sigma^S$-RNAP holoenzyme or pre-assembled Crl-$\sigma^S$–RNAP complex. In the assay, a Cy3 fluorophore, which is attached to the +2 position of the non-template strand of $\lambda P_R$ promoter DNA, is able to sense changes in the local environment upon promoter binding and unwinding (*Figure 4—figure supplement 1*). Such a technique has been employed to study the kinetics of RPo formation by *E. coli* $\sigma^{70}$-RNAP holoenzyme and *Mycobacterium tuberculosis* $\sigma^A$-RNAP holoenzyme (*Hubin et al., 2017*). The Cy3 fluorescence slowly reaches a plateau when $\sigma^S$-RNAP holoenzyme alone is mixed with promoter DNA; importantly, the presence of Crl substantially increases the plateau fluorescence (*Figure 4B*). The results suggest that Crl probably also functions at the step of $\sigma^S$-RPo formation.

To confirm above results, we repeated the stopped-flow fluorescence assay using an $\lambda P_R$ promoter derivative harboring 2-aminopurine (2-AP) at the −10 position of the template strand (*Figure 4—figure supplement 1*). The 2-AP is a better probe for RPo formation, because its fluorescence is quenched when 2-AP is base-paired and stacked in the context of duplex DNA, and is enhanced upon duplex DNA unwinding (*Sullivan et al., 1997*). The results in *Figure 4C* show that the 2-AP fluorescence increases slightly when pre-assembled $\sigma^S$-RNAP holoenzyme alone is rapidly mixed with promoter DNA; importantly, the presence of Crl greatly enhances the 2-AP fluorescence at plateau (*Figure 4C*), consistent with the above results using Cy3-fluorescence and confirming the effect of Crl on RPo formation.

To further visualize and confirm the effect of Crl on RPo formation, we employed a potassium permanganate ($KMnO_4$) footprinting approach to measure equilibrated RPo using the *OsmY* promoter, of which three thymines (−9 to −7, T-stretch) on the non-template strand of unwound transcription bubble would show permanganate reactivity upon RPo formation (red sequence in *Figure 4D*). The results show a weak but site-specific cleavage in the expected T-stretch site when the $\sigma^S$-RNAP holoenzyme is present in the reaction (*Figure 4D*, lane 2); and again the presence of Crl in the complex substantially increases the amount of RPo (*Figure 4D*, lane 3), consistent with a previous report (*Robbe-Saule et al., 2006*). On the basis of the evidence described above from two stopped-flow fluorescence experiments and the potassium permanganate footprinting assay, we conclude that Crl is able to promote $\sigma^S$–RPo formation.

In our structure, Crl uses its residue R51 to interact with D135 and E137 from the 'specificity loop' and secures the latter in a conformation that readily accepts the unwound nucleotide (*Figure 2—figure supplement 1D*) (*Liu et al., 2016*). Therefore, it is very likely that Crl may facilitate RPo formation by pre-organizing or stabilizing a key helix-loop-helix element ($\alpha4$-specificity loop-$\alpha5$) of $\sigma^S_2$, and the 'specificity loop' in turn interacts with the first unwound nucleotide to facilitate promoter unwinding or to stabilize the transcription bubble during the subsequent steps of transcription initiation (*Figure 4E–F*).

Evidence from HDX-MS also votes for such a notion. Residues from $\alpha4$ and $\alpha5$ that showed decreased HDX rate in the presence of Crl form an intimate interaction with the upstream junction of promoter DNA (*Figure 3G*). In the cryo-EM structure, residue R129 from $\alpha4$ may salt-bridge with the DNA phosphate backbone (*Figure 3G*). Meanwhile, residue Y145 from $\alpha5$ sits just underneath the base of the unwound $A_{-11}$ (*Figures 3G* and *4F*), and might facilitate the flipping of $A_{-11}$ by pi-stacking with its base. Together, such facts suggest that the rigidifying effect of Crl on helices $\alpha4$ and $\alpha5$ may promote not only the assembly of $\sigma^S$-RNAP holoenzyme but also the interaction between $\sigma^S$ and DNA, which in turn contribute to promoter unwinding or RPo stability.

Together, our biochemical data, cryo-EM structure and HDX-MS results clearly support a model in which Crl functions as a stabilizing chaperon to facilitate the assembly of $\sigma^S$-RNAP holoenzyme and RPo formation.

## Discussion

Crl was discovered as the specific transcription activator of the $\sigma^S$-RNAP holoenzyme about 20 years ago. A large collection of biochemical, biophysical and genetic data implies that Crl may activate transcription in an unprecedented manner. In this study, we determined a 3.80 Å cryo-EM structure

of *E. coli* Crl–TAC (*Figure 1*). The structure shows that Crl shields a large and otherwise solvent-exposed surface on $\sigma^S_2$ in the $\sigma^S$-RNAP holoenzyme and stabilizes the key 'specificity loop' of $\sigma^S_2$, but does not contact promoter DNA (*Figure 2*). Subsequent HDX-MS results recapitulate the interaction between Crl and the $\alpha2$–$\alpha3$ of $\sigma^S_2$ in solution, and further unravel that the stabilizing effect of Crl on $\sigma^S_2$ extends beyond the helix $\alpha2$–$\alpha3$ to $\alpha4$ and $\alpha5$ (*Figure 3C–F*). Considering the role of $\alpha4$ and $\alpha5$ of $\sigma^S_2$ in anchoring the RNAP-$\beta'$ subunit and promoter DNA, our cryo-EM structure and HDX-MS data thus point to a model in which binding of Crl stabilizes the conformation of $\sigma^S$ to promote the assembly of the $\sigma^S$-RNAP holoenzyme and the formation of RPo.

Most bacterial transcription factors activate gene expression by creating additional physical linkage between RNAP and the corresponding promoter DNA (*Browning and Busby, 2016*); whereas the eukaryotic transcription activators typically interact with both enhancer DNA element and mediator proteins, the latter bridging the transcription activators to transcription core machinery (*Jeronimo and Robert, 2017*). It is widely accepted that transcription factors activate transcription by enhancing the interaction between promoter DNA and the transcription machinery (*Browning and Busby, 2016*), so it will be intriguing to find out how Crl activates transcription as an unconventional transcription activator that does not contact promoter DNA.

Our structure and biochemical data reveal that Crl activates transcription by stabilizing the critical structural elements of $\sigma^S_2$: the 'RNAP-anchoring helix' and the 'specificity loop'. The effect on the 'specificity loop' is straightforward as Crl makes direct interactions with it. It is intriguing that the conformations of the 'RNAP-anchoring helix' are remotely restrained by Crl binding. Although it is not known how exactly the signal of Crl binding is allosterically transmitted to the 'RNAP-anchoring helix', we infer that Crl probably clamps the two helices ($\alpha2$ and $\alpha3$) of $\sigma^S_{1.2}$ and $\sigma^S_{2.1}$, reduces the internal motion of the helix bundle, and consequently stabilizes the conformer in a way that is competent for engagement to RNAP.

We point out that Cartagena and coworkers recently also reported a cryo-EM structure of a transcription activation complex comprising the *E. coli* RNAP core enzyme, *Salmonella enterica* serovar Typhimurium Crl and $\sigma^S$ (*Cartagena et al., 2019*). Our structure of *E. coli* Crl–TAC agrees well with this chimeric complex structure, and both structures show that Crl engages $\sigma^S$-RNAP through a large interface with $\sigma^S_2$ and a small interface with the RNAP core enzyme. *Cartagena et al. (2019)* proposed a 'tethering' mechanism by which Crl tethers $\sigma^S$ to the RNAP core enzyme and thus promotes $\sigma^S$-RNAP assembly. By contrast, we found that the interaction between Crl and the RNAP core enzyme only plays a partial role in $\sigma^S$-RNAP assembly. Instead, we demonstrate, using HDX-MS techniques, that Crl stabilizes critical structural elements of $\sigma^S_2$ — the 'RNAP-anchoring helix' and 'specificity loop' — to promote $\sigma^S$-RNAP assembly and to facilitate RPo formation.

Different from canonical bacterial transcription activators that interact with the RNAP-$\alpha$ subunit and/or region 4 of $\sigma$ factor, Crl specifically activates the $\sigma^S$-regulated genes by making interactions with region 2 of $\sigma^S$. A few recently discovered transcription activators, such as *C. crescentus* GcrA (*Fioravanti et al., 2013*; *Haakonsen et al., 2015*; *Wu et al., 2018*), *Mycobacterium tuberculosis* RbpA (*Bortoluzzi et al., 2013*; *Hu et al., 2012*; *Hubin et al., 2017*; *Hubin et al., 2015*), and *Chlamydia trachomatis* GrgA (*Bao et al., 2012*), also anchor RNAP through interactions with region 2 of primary $\sigma$ factors (*Figure 2—figure supplement 1E–G*), suggesting that the region 2 of $\sigma$ factors could also serve as a hub for docking various transcription activators. However, in contrast to Crl, these transcription activators contain additional domains that activate transcription by making essential interactions with promoter DNA in a sequence-independent (GrgA and RbpA) or sequence-dependent manner (GcrA) (*Bao et al., 2012*; *Fioravanti et al., 2013*; *Hubin et al., 2017*).

Collectively, we revealed here that Crl increases the transcriptional activity of $\sigma^S$-RNAP in a DNA contact-independent manner and through stabilizing the key structural elements of $\sigma^S$.

Our study provides the structural basis and molecular mechanism of an unprecedented example of transcription activation by *E. coli* Crl. The combined effect of Crl — facilitating assembly of $\sigma^S$-RNAP holoenzyme and assisting RPo formation — would help $\sigma^S$ to outcompete the housekeeping $\sigma$ and substantially increase the transcription activity of the $\sigma^S$-RNAP holoenzyme in the expression of stress-related genes. The unique DNA contact-independent mechanism also provides a new paradigm for bacterial transcription activation.

# Materials and methods

**Key resources table**

| Reagent type (species) or resource | Designation | Source or reference | Identifiers | Additional information |
|---|---|---|---|---|
| Gene (*Escherichia coli*) | *crl* | | Gene bank (7152411) | |
| Strain, strain background (*E. coli*) | BL21(DE3) | Novo protein, China | V211-01A | Chemically competent cells |
| Recombinant DNA reagent | pET-28a-TEV-Crl (plasmid) | This paper | | Protein expression vector for *E. coli* Crl |
| Recombinant DNA reagent | pET-28a-TEV-*Ecs*$^S$ (plasmid) | This paper | | Protein expression vector for *E. coli* s$^S$ |
| Recombinant DNA reagent | pET-28a-TEV-Crl (P21W) (plasmid) | This paper | | Protein expression vector for *E. coli* Crl (P21W) |
| Recombinant DNA reagent | pET-28a-TEV-Crl (Y22A) (plasmid) | This paper | | Protein expression vector for *E. coli* Crl (Y22A) |
| Recombinant DNA reagent | pET-28a-TEV-Crl (R24A) (plasmid) | This paper | | Protein expression vector for *E. coli* Crl (R24A) |
| Recombinant DNA reagent | pET-28a-TEV-Crl (L38A) (plasmid) | This paper | | Protein expression vector for *E. coli* Crl (L38A) |
| Recombinant DNA reagent | pET-28a-TEV-Crl (R51A) (plasmid) | This paper | | Protein expression vector for *E. coli* Crl (R51A) |
| Recombinant DNA reagent | pET-28a-TEV-Crl (F53A) (plasmid) | This paper | | Protein expression vector for *E. coli* Crl (F53A) |
| Recombinant DNA reagent | pET-28a-TEV-Crl (F76A) (plasmid) | This paper | | Protein expression vector for *E. coli* Crl (F76A) |
| Recombinant DNA reagent | pET-28a-TEV-Crl (ΔN-tail) (plasmid) | This paper | | Protein expression vector for *E. coli* Crl (Δ1–11) |
| Recombinant DNA reagent | pET-28a-TEV-Crl (ΔR-loop) (plasmid) | This paper | | pET-28a-TEV-Crl (ΔR-loop; residues 43–51 of Crl by a 'GSGS' linker) |
| Recombinant DNA reagent | pET-28a-TEV-Crl (ΔN-tail /ΔR-loop) (plasmid) | This paper | | Protein expression vector for *E. coli* Crl (ΔN-tail /ΔR-loop) |
| Recombinant DNA reagent | pET-28a-TEV-*Ec*σ$^S$(R81A) (plasmid) | This paper | | Protein expression vector for *E. coli* σ$^S$ (R81A) |
| Recombinant DNA reagent | pET-28a-TEV-*Ec*σ$^S$ (R82A) (plasmid) | This paper | | Protein expression vector for *E. coli* σ$^S$ (R82A) |
| Recombinant DNA reagent | pET-28a-TEV-*Ec*σ$^S$ (R85A) (plasmid) | This paper | | Protein expression vector for *E. coli* σ$^S$ (R85A) |
| Recombinant DNA reagent | pET-28a-TEV-*Ec*σ$^S$ (D135A) (plasmid) | This paper | | Protein expression vector for *E. coli* σ$^S$ (D135A) |
| Recombinant DNA reagent | pET-28a-TEV-*Ec*σ$^S$ (E137A) (plasmid) | This paper | | Protein expression vector for *E. coli*σ$^S$ (E137A) |
| Recombinant DNA reagent | pET-28a-TEV-*Ec*σ$^S$ (A239C) (plasmid) | This paper | | Protein expression vector for *E. coli* σ$^S$ (A239C) |

*Continued on next page*

*Continued*

| Reagent type (species) or resource | Designation | Source or reference | Identifiers | Additional information |
|---|---|---|---|---|
| Recombinant DNA reagent | pGADT7-Crl (plasmid) | This paper | | Plasmid for yeast two-hybrid; Crl is fused to Gal4-AD |
| Recombinant DNA reagent | pGADT7-Crl (P21W) (plasmid) | This paper | | Plasmid for yeast two-hybrid; Crl (P21W) is fused to Gal4-AD |
| Recombinant DNA reagent | pGADT7-Crl (Y22A) (plasmid) | This paper | | Plasmid for yeast two-hybrid; Crl (Y22A) is fused to Gal4-AD |
| Recombinant DNA reagent | pGADT7-Crl (R24A) (plasmid) | This paper | | Plasmid for yeast two-hybrid; Crl (R24A) is fused to Gal4-AD |
| Recombinant DNA reagent | pGADT7-Crl (E25A) (plasmid) | This paper | | Plasmid for yeast two-hybrid; Crl (E25A) is fused to Gal4-AD |
| Recombinant DNA reagent | pGADT7-Crl (L38A) (plasmid) | This paper | | Plasmid for yeast two-hybrid; Crl (L38A) is fused to Gal4-AD |
| Recombinant DNA reagent | pGADT7-Crl (V44A) (plasmid) | This paper | | Plasmid for yeast two-hybrid; Crl (V44A) is fused to Gal4-AD |
| Recombinant DNA reagent | pGADT7-Crl (K45A) (plasmid) | This paper | | Plasmid for yeast two-hybrid; Crl (K45A) is fused to Gal4-AD |
| Recombinant DNA reagent | pGADT7-Crl (R51A) (plasmid) | This paper | | Plasmid for yeast two-hybrid; Crl (R51A) is fused to Gal4-AD |
| Recombinant DNA reagent | pGADT7-Crl (F53A) (plasmid) | This paper | | Plasmid for yeast two-hybrid; Crl (F53A) is fused to Gal4-AD |
| Recombinant DNA reagent | pGADT7-Crl (F76A) (plasmid) | This paper | | Plasmid for yeast two-hybrid; Crl (F76A) is fused to Gal4-AD |
| Recombinant DNA reagent | pGADT7-Crl (ΔR-loop) (plasmid) | This paper | | Plasmid for yeast two-hybrid; Crl (ΔR-loop) is fused to Gal4-AD |
| Recombinant DNA reagent | pGBKT7-$\sigma^S_2$ (plasmid) | This paper | | Plasmid for yeast two-hybrid; $\sigma^S_2$(53–162) is fused to Gal4-BD |
| Recombinant DNA reagent | pGBKT7-$\sigma^S_2$ (Y78A) (plasmid) | This paper | | Plasmid for yeast two-hybrid; $\sigma^S_2$ (Y78A) is fused to Gal4-BD |
| Recombinant DNA reagent | pGBKT7-$\sigma^S_2$ (F79A) (plasmid) | This paper | | Plasmid for yeast two-hybrid; $\sigma^S_2$ (F79A) is fused to Gal4-BD |
| Recombinant DNA reagent | pGBKT7-$\sigma^S_2$ (R81A) (plasmid) | This paper | | Plasmid for yeast two-hybrid; $\sigma^S_2$ (R81A) is fused to Gal4-BD |
| Recombinant DNA reagent | pGBKT7-$\sigma^S_2$ (R82A) (plasmid) | This paper | | Plasmid for yeast two-hybrid; $\sigma^S_2$ (R82A) is fused to Gal4-BD |
| Recombinant DNA reagent | pGBKT7-$\sigma^S_2$ (R85A) (plasmid) | This paper | | Plasmid for yeast two-hybrid; $\sigma^S_2$ (R85A) is fused to Gal4-BD |
| Recombinant DNA reagent | pGBKT7-$\sigma^S_2$ (R93A) (plasmid) | This paper | | Plasmid for yeast two-hybrid; $\sigma^S_2$ (R93A) is fused to Gal4-BD |

*Continued on next page*

*Continued*

| Reagent type (species) or resource | Designation | Source or reference | Identifiers | Additional information |
|---|---|---|---|---|
| Recombinant DNA reagent | pGBKT7-$\sigma^S_2$ (D135A) (plasmid) | This paper | | Plasmid for yeast two-hybrid; $\sigma^S_2$ (D135A) is fused to Gal4-BD |
| Recombinant DNA reagent | pGBKT7-$\sigma^S_2$ (E137A) (plasmid) | This paper | | Plasmid for yeast two-hybrid; $\sigma^S_2$ (E137A) is fused to Gal4-BD |
| Recombinant DNA reagent | pEasyT-PosmY (plasmid) | This paper | | Plasmid containing P*osmY* |
| Commercial assay or kit | Ezmax one-step cloning kit | Tolo Bio-tech, China | Cat#24305–01 | |
| Other | Ni-NTA agarose | smart-lifesciences, China | Cat#SA004100 | |
| Other | C-flat CF-1.2/1.3 400 mesh | Electron Microscopy Sciences | Cat#CF413-100 | |

## Plasmid construction

DNA fragments containing *E. coli crl* and *rpoS* were amplified from *E. coli* genomic DNA and cloned into pET28a-TEV using the Ezmax one-step cloning kit (Tolo Bio-tech, China). Point mutations of *crl* or *rpoS* were generated through site-directed mutagenesis (Transgen biotech, Inc). pET-28a-TEV-Crl (ΔR-loop) was constructed by replacing residues 43–51 of Crl with a 'GSGS' linker. pGADT7-Crl and pGBKT7-$\sigma^S_2$ were constructed using the Gateway LR clonase II (Invitrogen, Inc). The derivatives of pGADT7-Crl and pGBKT7-$\sigma^S_2$ were generated through site-directed mutagenesis.

## Protein preparation

The *Ec* RNAP core enzyme was overexpressed and purified from *E. coli* BL21(DE3) carrying p*Ec*ABC and pCDF-*Ec* rpoZ as described previously (*Hudson et al., 2009*).

The *Ec* Crl was overexpressed in *E. coli* BL21(DE3) cells (Novo protein, Inc) carrying pET28a-TEV-Crl. Protein expression was induced with 0.5 mM isopropyl β-d-1-thiogalactopyranoside (IPTG) at 18° C for 14 hr when $OD_{600}$ reached to 0.6–0.8. Cell pellet was lysed in lysis buffer (50 mM Tris-HCl [pH 7.7], 500 mM NaCl, 5% (v/v) glycerol, 5 mM β-mercaptoethanol, and protease inhibitor cocktail [Bio-make.cn. Inc]) using an Avestin EmulsiFlex-C3 cell disrupter (Avestin, Inc). The lysate was centrifuged (16,000 g; 50 min; 4°C) and the supernatant was loaded onto a 2 ml column packed with Ni-NTA agarose (smart-lifesciences, Inc). The proteins bound on resin were washed by the lysis buffer containing 20 mM imidazole and eluted with the lysis buffer containing 300 mM imidazole. The eluted fractions were mixed with tobacco etch virus (TEV) protease and dialyzed against 20 mM Tris-HCl (pH 7.7), 50 mM NaCl, and 1 mM DTT. The sample was reloaded onto a Ni-NTA column to remove his tag. The Crl was further purified on a Q HP column (HiPrep Q HP 16/10, GE healthcare Life Sciences) with a salt gradient of buffer A (20 mM Tris-HCl [pH 7.7], 50 mM NaCl, and 1 mM DTT) and buffer B (20 mM Tris-HCl [pH 7.7], 1 M NaCl, and 1 mM DTT). The fractions containing target proteins were collected, concentrated, and stored at −80°C.

The Crl and $\sigma^S$ derivatives were prepared by the same procedure.

## Nucleic-acid scaffolds

For Cryo-EM study, the DNA sequence of upstream half promoter (−36 to −7) was chosen on the basis of the nuclei-acid scaffold for obtaining the crystal structure of *E. coli* $\sigma^S$–RPo (*Liu et al., 2016*); whereas the DNA sequence of the downstream half promoter (−6 to +15) was chosen on the basis of the nuclei-acid scaffold for obtaining the crystal structure of $\sigma^A$–RPo (*Zhang et al., 2012*). The nucleic-acid scaffold was prepared by mixing synthetic non-template DNA, template DNA, and RNA at molar ratio of 1:1.2:1.5 and subjected to an annealing procedure (95°C, 5 min followed by 2°C-step cooling to 25°C) in annealing buffer (5 mM Tris-HCl [pH 8.0], 200 mM NaCl, and 10 mM $MgCl_2$).

## Complex reconstitution of *E. coli* Crl–TAC

The Crl–σ$^S$ binary complex was prepared by incubating Crl and σ$^S$ at a molar ratio of 2:1 and purified by using a Superdex 75 gel filtration column (GE Healthcare). The *E. coli* Crl–TAC was assembled by directly incubating the RNAP core enzyme, the Crl–σ$^S$ binary complex, and the nucleic-acid scaffold at a molar ratio of 1:4:4 at 4°C overnight. The mixture was loaded onto a Superdex 200 gel filtration column (GE Healthcare) and eluted with 10 mM HEPES [pH 7.5], 50 mM KCl, 5 mM MgCl$_2$, and 3 mM DTT. Fractions containing *E. coli* Crl–TAC were collected and concentrated to ~12 mg/ml.

## Cryo-EM structure determination of *E. coli* Crl–TAC

The *E. coli* Crl–TAC sample was freshly prepared as described above and mixed with CHAPSO (Hampton Research, Inc) to a final concentration 8 mM prior to grid preparation. About 4 µL of the complex sample was applied onto the glow-discharged C-flat CF-1.2/1.3 400 mesh holey carbon grids (Electron Microscopy Sciences) and the grid was plunge-frozen in liquid ethane using a Vitrobot Mark IV (FEI) with 95% chamber humidity at 10°C.

The data were collected on a 300 keV Titan Krios (FEI) equipped with a K2 Summit direct electron detector (Gatan). A total of 3290 images of Crl–TAC were recorded using the Serial EM (*Mastronarde, 2005*) in super-resolution counting mode with a pixel size of 0.507 Å, and a dose rate of 6.7 electrons/pixel/s. Movies were recorded at 250 ms/frame for 8 s (32 frames total) and defocus range was varied between 2.0 µm and 2.5 µm. Frames in individual movies were aligned using MotionCor2 (*Zheng et al., 2017*), and Contrast-transfer-function estimations were performed using CTFFIND (*Rohou and Grigorieff, 2015*). About 1160 particles were picked and subjected to 2D classification in RELION 3.0 (*Fernandez-Leiro and Scheres, 2017*). The resulting distinct two-dimensional classes were served as templates and a total of 315,977 particles for Crl–TAC were picked out. The resulting particles were manually inspected and subjected to 2D classification in RELION 3.0 by specifying 100 classes (*Zivanov et al., 2018*). Poorly populated classes were removed. We used a 50 Å low-pass-filtered map calculated from structure of *E. coli* σ$^S$–TIC (*Kang et al., 2017*) (PDB: 5IPL) as the starting reference model for 3D classification. The final maps were obtained through 3D auto-refinement, CTF-refinement, Bayesian polishing, and post-processing in RELION 3.0 (*Figure 1—figure supplement 2*). Gold-standard Fourier-shell-correlation analysis (FSC) (*Henderson et al., 2012*) indicated a mean map resolution of 3.80 Å for *Ec* Crl–TAC.

The crystal structure of *E. coli* σ$^S$–TIC (*Kang et al., 2017*)(PDB: 5IPL) and the crystal structure of *P. mirabilis* Crl (*Cavaliere et al., 2014*) (PDB: 4Q11) were manually fit into the cryo-EM density map using Chimera (*Pettersen et al., 2004*). Rigid body and real-space refinement was performed in Coot (*Emsley and Cowtan, 2004*) and Phenix (*Adams et al., 2010*).

## Hydrogen–deuterium exchange mass spectrometry (HDX-MS) of σ$^S$

The HDX-MS was performed as recommended in *Masson et al. (2019)*. Amide hydrogen exchange of σ$^S$ alone was started by diluting 3 µl protein sample at 19 µM into 27 µl D$_2$O buffer (20 mM Tris [pH 7.7], 150 mM NaCl, 1 mM tris(2-carboxyethyl)phosphine [TCEP]) at 20°C. At different time points (0 s, 10 s, 60 s, 300 s and 900 s), the labeling reaction was quenched by the addition of chilled quench buffer (200 mM KH$_2$PO$_4$/K$_2$HPO$_4$ [pH 2.2]) and the reaction mixture was immediately frozen in liquid nitrogen. For the HDX-MS of σ$^S$ in the presence of Crl, 50 µL σ$^S$ at 34 µM were first mixed with 40 µl Crl at 176 µM and incubated at room temperature for 1 hr. 3 µL mixture was then labeled by adding 27 µL D$_2$O buffer before being quenched at the above time points and flash-frozen. All frozen samples were stored at −80°C until analysis.

The thawed samples were immediately injected into an HPLC-MS (Agilent 1100) system equipped with in-line peptic digestion and desalting. The desalted digests were then separated with a Hypersil Gold C18 analytical column (ThermoFisher) over a 19 min gradient and directly analyzed with an Orbitrap Fusion mass spectrometer (ThermoFisher). The HPLC system was extensively cleaned with blank injections between samples to minimize any carryover. Peptides identification was performed by tandem MS/MS in the orbi/orbi mode. All peptides were identified using the Proteome Discoverer Software (ThermoFisher). We carried out the initial analysis of the peptide centroids with HD-Examiner v2.3 (Sierra Analytics) and then manually verified every peptide to check retention time, charge state, m/z range and the presence of overlapping peptides. The peptide coverage of σ$^S$ were found to be 95.2% and the relative deuteration levels (%D) of each peptide were automatically

calculated by HD-Examiner with the assumption that a fully deuterated sample retains 90% D in current LC setting.

## Yeast two-hybrid assay

The GAL4-based yeast two-hybrid system (MATECHMAKER GAL4 two-hybrid system3, Clontech Laboratories, Inc) was used to analyze the protein-protein interaction according to the standard procedure. Briefly, wild-type or derivatives of $Ec$Crl and $Ec\sigma^S_2$ were cloned into prey vector pGADT7 and the bait vector pGBKT7, respectively. The bait and prey vectors were transformed into Y187 and AH109 yeast cells, respectively. The haploid colonies of Y187 were mated with haploid prey colonies of AH109 in the YPDA medium for 24 hr, and the diploid yeast cells containing both bait and prey vectors were selected on SD (–Leu, –Trp) plates at 30℃ for 48 hr. The colonies were inoculated into SD (–Leu, –Trp) medium and cultured at 30℃ for 24 hr. The resulting cell suspensions with a series of dilution were spotted onto SD (–Leu, –Trp) and SD (–Ade, –His, –Leu, –Trp) plates, incubated at 30℃ for 4–5 days. Positive colonies appear after 3 days on SD (–Ade, –His, –Leu, –Trp) plates and the plate images were taken 5–6 days after plating.

## Fluorescence labeling

The E. coli $\sigma^S$(A239C) was labeled with fluorescein at residues C239. The labeling reaction mixture (2 mL) containing $\sigma^S$ (0.07 mM) and fluorescein-5-maleimide (0.7 mM; Thermo Scientific, Inc) in 10 mM Tris-HCl, pH 7.7), 100 mM NaCl, and 1% glycerol was incubated overnight at 4℃. The reaction was terminated by addition of 2 μL 1M DTT, and loaded onto a 5 mL PD-10 desalting column (Biorad, Inc). The fractions containing labeled protein was pooled and concentrated to 3 mg/mL. The labeling efficiency is estimated at ~70%.

## Fluorescence polarization

The reaction mixtures (100 μL) contain the fluorescein-labeled $\sigma^S$ (4 nM; final concentration) with or without Crl (1 μM; final concentration) in FP buffer (10 mM Tris-HCl [pH 7.9], 300 mM NaCl, 1 mM DTT, 1% glycerol, and 0.025% Tween-20) were incubated for 10 min at room temperature. RNAP core enzyme (0 nM to 1024 nM; final concentration) was added and incubated for 10 min at room temperature. The FP signals were measured using a plate reader (SPARK, TECAN Inc) equipped with an excitation filter of 485/20 nm and an emission filter of 520/20 nm. The data were plotted in SigmaPlot (Systat software, Inc) and the dissociation constant $K$d was estimated by fitting the data to the following equation:

$$\mathrm{F} = \mathrm{B}[S]/(K\mathrm{d}+[S])+F_0$$

where F is the FP signal at a given concentration of RNAP, $F_0$ is the FP signal in the absence of RNAP, [S] is the concentration of RNAP, and B is an unconstrained constant.

## Stopped-flow assay

The stopped-flow assay was performed essentially as in *Feklistov et al. (2017)*. The Cy3-λP$_R$ promoter (−60 to + 53) with Cy3-amido-dT at position +2 of the non-template strand was prepared by PCR extending (−60 to + 53; *Figure 4—figure supplement 1A*). To monitor promoter melting, 60 μL pre-assembled $Ec$ $\sigma^s$-RNAP holoenzyme or pre-assembled $Ec$ $\sigma^s$-Crl-RNAP holoenzyme (200 nM; final concentration) and 60 μL Cy3-λP$_R$ promoter DNA (20 nM; final concentration) in 10 mM Tris-HCl [pH 7.7], 20 mM NaCl, 10 mM MgCl$_2$, and 1 mM DTT were rapidly mixed, and the change of fluorescence was monitored in real time by a stopped-flow instrument (SX20, Applied Photophysics Ltd, UK) using an excitation wavelength of 519 nm (slit width = 9.3 nm) and a long-pass emission filter (570 nm).

The 2AP-λP$_R$ promoter (−60 to + 53) with 2-amido purine at position −10 of the template strand was prepared by PCR extending (*Figure 4—figure supplement 1B*). To monitor the promoter melting, 60 μL pre-assembled $Ec$ $\sigma^s$-RNAP holoenzyme or pre-assembled $Ec$ $\sigma^s$-Crl-RNAP holoenzyme (400 nM; final concentration) and 60 μL Cy3-λ P$_R$ promoter DNA (100 nM; final concentration) in 10 mM Tris-HCl [pH 7.7], 20 mM NaCl, 10 mM MgCl$_2$, and 1 mM DTT were rapidly mixed, and the change of 2-AP fluorescence was monitored in real time by a stopped-flow instrument (SX20,

Applied Photophysics Ltd, UK) using an excitation wavelength of 309 nm (slit width 9.3 nm) and a long-pass emission filter (360 nm).

## In vitro transcription assay

The *osmY* promoter with a tR2 terminator was prepared by PCR amplification of the *osmY* promoter region (−107/+50 relative to the transcription start site) from *E. coli* genomic DNA using primers (forward primer: 5′–TTCCCTTCCTTATTAGCCGCTT−3′; reverse primer: 5′–AAATAAAAAGGCC TGCGATTACCAGCAGGCCCTGATATCTACGCATTGAACGG−3′). All of the in vitro transcription reactions were performed in transcription buffer (40 mM Tris-HCl [pH 7.9], 75 mM KCl, 5 mM MgCl$_2$, 12.5% glycerol, and 2.5 mM DTT) in a 20 µl reaction mixture.

To study the effect of overall transcription activation by Crl, Crl (500 nM; final concentration) was pre-incubated with σ$^S$ (200 nM; final concentration) for 10 min at 30°C in transcription buffer prior to addition of RNAP core enzyme (100 nM; final concentration). Promoter DNA (100 nM; final concentration) was subsequently added and incubated for 10 min. RNA synthesis was allowed by addition of NTP mixture (30 µM ATP, 30 µM CTP, 30 µM GTP, 30 µM [α-$^{32}$P]UTP [0.04 Bq/fmol]; final concentration) for 15 min at 30°C. The reactions were terminated by adding 5 µL loading buffer (8 M urea, 20 mM EDTA, 0.025% xylene cyanol, and 0.025% bromophenol blue), boiled for 2 min, and cooled down in ice for 5 min.

To study the transcription activation of pre-assembled σ$^s$-RNAP holoenzyme by Crl, reaction mixture containing pre-assembled *E. coli* σ$^s$-RNAP (200 nM; final concentration) and Crl (0 nM to 1600 nM; final concentration) were incubated in transcription buffer for 10 min at 30°C, then promoter DNA (200 nM; final concentration) was added before the mixture was incubated for 10 min at 30°C for open complex formation. RNA synthesis was allowed by the addition of NTP mixture (30 µM ATP, 30 µM CTP, 30 µM GTP, 30 µM [α-$^{32}$P]UTP [0.04 Bq/fmol] for each; final concentration) for 15 min at 30°C. The reactions were terminated by adding 5 µL loading buffer (8 M urea, 20 mM EDTA, 0.025% xylene cyanol, and 0.025% bromophenol blue), boiled for 2 min, and cooled down in ice for 5 min. The RNA transcripts were separated by 15% urea-polyacrylamide slab gels (19:1 acrylamide/ bisacrylamide) in 90 mM Tris-borate (pH 8.0) and 0.2 mM EDTA, and analyzed by storage-phosphor scanning (Typhoon; GE Healthcare, Inc).

## Potassium permanganate KMnO$_4$ footprinting assay

The *osmY* promoter dsDNA with $^{32}$P-labeled at the 5′-end of non-template strand for the footprinting assay was prepared as follows. The pEasyT-P*osmY* was constructed by ligation of pEasy-Blunt and a dsDNA fragment containing *osmY* promoter DNA (−66/+27) followed by an *EcoRI* site, which was prepared by annealing synthetic non-template and template oligodeoxynucleotides (*osmY*-NT: 5′- CACTTTTGCTTATGTTTTCGCTGATATCCCGAGCGGTTTCAAAATTGTGATCTATATTTAA-CAAAGTGATGACATTTCTGACGGCGTTAAATAGAATTC-3′; *osmY*-T: 5′- GAATTCTATTTAACGCCG TCAGAAATGTCATCACTTTGTTAAATATAGATCACAATTTTGAAACCGCTCGGGATATCAGC-GAAAACATAAGCAAAAGTG-3′). Subsequently, a dsDNA fragment of 520 bp containing *osmY* promoter (amplified from pEasyT-P*osmY* using forward primer 5′-CACTTTTGCTTATGTTTTC-3′ and reverse primer 5′-ACCCTAATCAAGTTTTTTGGGGTC-3′) was labeled with γ-$^{32}$P-ATP (PerkinElmer, Inc) and T4 Polynucleotide Kinase (NEB, Inc) at 37°C for 1 hr, and purified using illustra MicroSpin G-25 columns (GE Healthcare, Inc). The labeled dsDNA was digested by *EcoRI* at 37°C for 1 hr, and separated by native PAGE electrophoresis. The final *osmY* promoter dsDNA with $^{32}$P-labeled at the 5′-end of the non-template strand was purified from the gel and quantified.

The reaction mixture (25 µl) containing pre-assembled *E. coli* σ$^s$-RNAP (200 nM; final concentration) or *E. coli* Crl-σ$^s$-RNAP (200 nM; final concentration), $^{32}$P-labeled *osmY* promoter DNA (50 nM; final concentration), 10 mM Tris Tris-HCl (pH 7.9), 100 mM NaCl, and 5 mM MgCl$_2$ was incubated at room temperature for 15 min. Subsequently, 1 µl potassium permanganate (KMnO$_4$, 2.5 mM; final concentration) was added and the mixture was incubated at 37°C for 2 min. The reaction was stopped by adding 24 µl β-mercaptoethanol (250 mM; final concentration) and incubating for 15 s. DNA was precipitated and rinsed with ethanol, resuspended in 100 µl 1M piperidine, and heated at 90°C for 30 min. After piperidine cleavage, the DNA was precipitated with 100% ethanol, pelleted and washed with 75% ethanol. The pellet was air-dried, resuspended in 20 µl loading buffer (8 M

urea, 20 mM EDTA, 0.025% xylene cyanol, and 0.025% bromophenol blue) and boiled for 2 min before loading onto a 15% urea-polyacrylamide slab gel.

The A+G marker was prepared essentially as in *Ross and Gourse (2009)*. Briefly, 12 µl $^{32}$P-labeled P*osmY* (20 nM; final concentration) was incubated with 50 µl formic acid at room temperature for 7 min. The depurinated DNA was precipitated and rinsed with 100% ethanol. The pellet was then resuspended in 100 µl 1 M piperidine and heated at 90℃ for 30 min. The cleaved DNA was precipitated with 100% (v/v) ethanol, washed with 75% ethanol, air-dried, and resuspended in 20 µl loading buffer. The sample was then boiled for 2 min and loaded onto a 15% urea-polyacrylamide slab gel.

## Quantification and statistical analysis
All biochemical assays were performed at least three times independently. Data were analyzed with SigmaPlot 10.0 (Systat Software Inc).

## Acknowledgements
We thank Dr Zhaocai Zhou for the generous gift of pET28a-TEV plasmid, Dr Shenghai Chang at the cryo-EM center of Zhejiang University for assistance with grid preparation and data collection, and Dr Chun Tang for valuable discussion. We thank the state key laboratory of bioorganic and natural products chemistry at Shanghai Institute of Organic Chemistry at CAS for sharing the stopped-flow fluorescence spectrometer.

## Additional information

### Funding

| Funder | Grant reference number | Author |
| --- | --- | --- |
| Ministry of Science and Technology of the People's Republic of China | 2018YFA0900701 | Yu Zhang |
| Chinese Academy of Sciences | XDB29020000 | Yu Zhang |
| Chinese Academy of Sciences | QYZDB-SSW-SMC005 | Yu Zhang |
| National Natural Science Foundation of China | 31822001 | Yu Zhang |
| National Natural Science Foundation of China | 31870047 | Yu Zhang |

The funders had no role in study design, data collection and interpretation, or the decision to submit the work for publication.

### Author contributions
Juncao Xu, Kaijie Cui, Liqiang Shen, Jing Shi, Lingting Li, Linlin You, Chengli Fang, Investigation; Guoping Zhao, Yu Feng, Resources, Supervision, Project administration, Writing—review and editing; Bei Yang, Conceptualization, Resources, Supervision, Project administration, Writing—review and editing; Yu Zhang, Conceptualization, Resources, Supervision, Funding acquisition, Writing—original draft, Project administration, Writing—review and editing

### Author ORCIDs
Bei Yang https://orcid.org/0000-0001-5389-3859
Yu Zhang https://orcid.org/0000-0002-1778-8389

### Decision letter and Author response
Decision letter https://doi.org/10.7554/eLife.50928.sa1
Author response https://doi.org/10.7554/eLife.50928.sa2

## Additional files

### Supplementary files

- Supplementary file 1. The statistics of the cryo-EM structure of *E. coli* Crl–TAC.
- Supplementary file 2. The estimated equilibrium dissociation constants from *Figure 3B* of RNAP and σ$^S$ in the presence of wild-type or mutant Crl.
- Source code 1. The map for structure of Ec-Crl–TAC.
- Transparent reporting form

### Data availability

The cryo-EM map of *E. coli* Crl-TAC has been deposited to the EMDB under accession number EMD-0700. The coordinate of *E. coli* Crl-TAC has been deposited to the PDB under accession number 6KJ6.

The following datasets were generated:

| Author(s) | Year | Dataset title | Dataset URL | Database and Identifier |
|---|---|---|---|---|
| Juncao Xu, Yu Zhang | 2019 | Cryo-EM structure of Escherichia coli Crl transcription activation complex | https://www.rcsb.org/structure/6KJ6 | Protein Data Bank, 6KJ6 |
| Juncao Xu, Yu Zhang | 2019 | Cryo-EM structure of Escherichia coli Crl transcription activation complex | https://www.ebi.ac.uk/pdbe/entry/emdb/EMD-0700 | Electron Microscopy Data Bank, EMD-0700 |

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
