## [Decision Letter]

**Acceptance summary:**

Stationary σ factor (σ^S^) transcribe genes in *E. coli* cells responsible for stress responses and antibiotic resistances. The σ^S^-dependent transcription is activated by the σ^S^-specific transcription activator Crl, although previous studies indicated that the Crl doesn't contact with promoter DNA. In this work, Zhang and colleagues build on previous studies reporting the cryo-EM structure of RNAP σ^S^-holoenzyme (Eσ^S^) and promoter DNA complex in the presence of Crl at 3.8 Å resolution, which confirmed lack of contact between Crl and DNA. Authors claimed that the structure together with biochemical experiments propose a new mechanism of transcription activation, however, this conclusion is not well justified in an original manuscript. In the revised manuscript, the editors judge that you have addressed all of the issues raised by reviewers and we are delighted to be able to publish your work in *eLife*. The work is of interest to both structural and microbiologists.

**Decision letter after peer review:**

Thank you for submitting your article "Crl activates transcription by stabilizing active conformation of the master stress transcription initiation factor" for consideration by *eLife*. Your article has been reviewed by three peer reviewers, including Katsuhiko Murakami as the Guest Editor and Reviewer #1, and the evaluation has been overseen by Gisela Storz as the Senior Editor.

The reviewers have discussed the reviews with one another and the Guest Editor has drafted this decision to help you prepare a revised submission.

Summary:

The reviewers find that the paper proposes a new and an interesting mechanism of the Crl-dependent transcription activation, which is specific to the stationary phase σ factor (σ^S^) RNA polymerase holoenzyme in *E. coli*. The structure of Crl bound σ^S^ holoenzyme and promoter DNA complex was determined by cryo-EM without any issue (based on the submitted validation file, structure coordinate and EM map) and it clearly showed that the Crl mainly interacts with the σ domain 2 with some minor contacts with β-prime subunit (clamp head) but no contact with DNA. The key residues and regions involved in these interactions were derivatized and analyzed their interactions by yeast two-hybrid, in vitro transcription and fluorescence assays. Additional data of hydrogen-deuterium exchange mass spectrometry are also corroborating the structural revelations. However, this conclusion is not well justified due to lack of control data in the biochemical experiments, as described in the essential revisions.

The findings presented here are consistent with the recently published the structure of a similar complex reported by the Darst group (Cartagena et al., 2019). Despite the overlap, reviewers consider that this manuscript is suitable for publication once the comments below are addressed satisfactorily, and the Discussion incorporates a comparison highlighting differences or common findings in the recent paper by Cartagena et al.

Essential revisions:

The reviewers raise concerns that must be adequately addressed before the paper can be accepted.

1) Authors proposed that the interaction between the Crl R-loop (R51) and the sigmaS specificity loop (D135 and E137) facilitates the DNA unwinding by the σ region2 due to stabilization of the specificity loop conformation, based on the observation from the structure, the in vitro transcription and the Cy3 fluorescence assays with the site-directed derivatives of sigmaS and Crl. However, the same derivatives influenced the binding of Crl to sigmaS (yeast two hybrid assay, Figure 2E). Clarification of the roles of R-loop and specificity loop in binding versus activity is essential. Authors purified the RNAP and Crl complex by the gel-filtration for only the cryo-EM sample preparation but simply mixing RNAP and Crl for other assays. Authors should purify some key complexes by gel-filtration to provide evidence that these derivatives are only affecting functions but not the complex formation.

2) One of the major conclusions of this paper is that Crl promotes DNA unwinding by the σ domain 2 therefore facilitates the open complex formation. The experiment presented in this study, monitoring the fluorescence signal from Cy3 attached on the promoter DNA, can monitor changing environment around it, but do not directly probe the DNA unwinding. Authors consider presenting other methods (such as KMnO_4_ footprinting or 2-amino purine fluorescence) to directly probe for DNA unwinding, hallmarks of the open complex formation.

---

## [Author Response]

[…] The findings presented here are consistent with the recently published the structure of a similar complex reported by the Darst group (Cartagena et al., 2019). Despite the overlap, reviewers consider that this manuscript is suitable for publication once the comments below are addressed satisfactorily, and the Discussion incorporates a comparison highlighting differences or common findings in the recent paper by Cartagena et al.

We have added a paragraph in the Discussion section to compare the recent paper by Cartagena et al.

**“**We point out Cartagena and coworkers recently have also reported a cryo-EM structure of transcription activation complex comprising *E. coli* RNAP core enzyme, *Salmonella enterica* serovar Typhimurium Crl and σ^S^ (Cartagena et al., 2019). […] Instead, we demonstrate, using HDX-MS techniques, that Crl stabilizes critical structural elements of σ^S^_2_ – the “RNAP-anchoring helix” and “specificity loop” – to promoter σ^S^-RNAP assembly and facilitate RPo formation.**”**

Essential revisions:The reviewers raise concerns that must be adequately addressed before the paper can be accepted.1) Authors proposed that the interaction between the Crl R-loop (R51) and the sigmaS specificity loop (D135 and E137) facilitates the DNA unwinding by the σ region2 due to stabilization of the specificity loop conformation, based on the observation from the structure, the in vitro transcription and the Cy3 fluorescence assays with the site-directed derivatives of sigmaS and Crl. However, the same derivatives influenced the binding of Crl to sigmaS (yeast two hybrid assay, Figure 2E). Clarification of the roles of R-loop and specificity loop in binding versus activity is essential. Authors purified the RNAP and Crl complex by the gel-filtration for only the cryo-EM sample preparation but simply mixing RNAP and Crl for other assays. Authors should purify some key complexes by gel-filtration to provide evidence that these derivatives are only affecting functions but not the complex formation.

We thank the reviewers for the critical comments. R51A was the only mutant used for exploring the potential effect of Crl on RPo formation, as R51 is the only residue making direct interaction with structure segments that are closely related to promoter unwinding. As suggested by the reviewer, we sought to prepare the complex of RNAP-σ^S^-Crl(R51A) by incubating RNAP, σ^S^, and Crl(R51A) followed by gel-filtration separation of the complex, however, the RNAP-σ^S^-Crl(R51A) complex didn’t survive through gel-filtration and thus is not possible to perform the stopped-flow experiment using the pre-formed RNAP-σ^S^-Crl(R51A) complex. Although we have added much excess of Crl (WT of R51A) over RNAP-σ^S^ holoenzyme (1 µM versus 200 nM) in the stopped-flow fluorescence assay, it is still difficult to dissect the contributions of such interaction to Crl-σ^S^ binding and RPo formation. Therefore, we decide to remove the curve of R51A in Figure 4B and the corresponding text. We modified the words in the subsection “Crl stabilizes σ^S^_2_ to assist in promoter unwinding RPo formation” as follows,

“Therefore, it is possible that Crl may contribute in the promoter unwinding step by preorganizing the “specificity loop” in its active conformation, or in the RPo stability by stabilizing the “specificity loop” for securing the unwounded nontemplate A_-11_ nucleotide (Figure 4E-F).”

The reviewer also suggested using pre-assembled Crl-σ^S^-RNAP complex to explore the effect of Crl on RPo formation. We repeated the stopped-flow Cy3-fluorescence experiment using pre-assembled Crl-σ^S^-RNAP and the new results in Figure 4B shows even more pronounced RPo-promoting effect of Crl than the previous data using mixture of RNAP and Crl.

2) One of the major conclusions of this paper is that Crl promotes DNA unwinding by the σ domain 2 therefore facilitates the open complex formation. The experiment presented in this study, monitoring the fluorescence signal from Cy3 attached on the promoter DNA, can monitor changing environment around it, but do not directly probe the DNA unwinding. Authors consider presenting other methods (such as KMnO_4_ footprinting or 2-amino purine fluorescence) to directly probe for DNA unwinding, hallmarks of the open complex formation.

Thanks for the comment. We have performed two additional experiments to demonstrate the effect of Crl on RPo formation. In the first experiment, we repeated stopped-flow fluorescence experiment using an lP_R_ promoter derivative harboring 2-aminopurine (2-AP) at the -10 position of template strand. In the second experiment, we performed KMnO_4_ footprinting assay using *OsmY* promoter. In both assays, we used pre-assembled Crl-σ^S^ RNAP complex and σ^S^-RNAP for direct comparison as suggested by reviewers. Results from both experiments confirm that Crl is able to promoter RPo formation. The results have been incorporated into Figure 4B-C and the following description has been included into the text.

“To confirm above results, we repeated the stopped-flow fluorescence assay using an lP_R_ promoter derivative harboring 2-aminopurine (2-AP) at the -10 position of template strand[…] Based on above evidence from two stopped-flow fluorescence experiments and potassium permanganate footprinting assay, we conclude that Crl is able to facilitate σ^S^-RPo formation.”